# Nucleolar and Ribosomal DNA Structure under Stress: Yeast Lessons for Aging and Cancer

**DOI:** 10.3390/cells8080779

**Published:** 2019-07-26

**Authors:** Emiliano Matos-Perdomo, Félix Machín

**Affiliations:** 1Unidad de Investigación, Hospital Universitario Nuestra Señora de Candelaria, 38010 Santa Cruz de Tenerife, Spain; 2Escuela de Doctorado y Estudios de Postgrado, Universidad de La Laguna, 38200 Tenerife, Spain; 3Instituto de Tecnologías Biomédicas, Universidad de La Laguna, 38200 Tenerife, Spain; 4Facultad de Ciencias de la Salud, Universidad Fernando Pessoa Canarias, 35450 Santa María de Guía, Gran Canaria, Spain

**Keywords:** yeast nucleolus, yeast rDNA, nucleolar stress, TORC1, ribosome biogenesis, sirtuins, nucleolar condensation, aging, cancer

## Abstract

Once thought a mere ribosome factory, the nucleolus has been viewed in recent years as an extremely sensitive gauge of diverse cellular stresses. Emerging concepts in nucleolar biology include the nucleolar stress response (NSR), whereby a series of cell insults have a special impact on the nucleolus. These insults include, among others, ultra-violet radiation (UV), nutrient deprivation, hypoxia and thermal stress. While these stresses might influence nucleolar biology directly or indirectly, other perturbances whose origin resides in the nucleolar biology also trigger nucleolar and systemic stress responses. Among the latter, we find mutations in nucleolar and ribosomal proteins, ribosomal RNA (rRNA) processing inhibitors and ribosomal DNA (rDNA) transcription inhibition. The p53 protein also mediates NSR, leading ultimately to cell cycle arrest, apoptosis, senescence or differentiation. Hence, NSR is gaining importance in cancer biology. The nucleolar size and ribosome biogenesis, and how they connect with the Target of Rapamycin (TOR) signalling pathway, are also becoming important in the biology of aging and cancer. Simple model organisms like the budding yeast *Saccharomyces cerevisiae*, easy to manipulate genetically, are useful in order to study nucleolar and rDNA structure and their relationship with stress. In this review, we summarize the most important findings related to this topic.

## 1. Introduction

A membraneless compartment inside the nucleus of eukaryotic cells was observed for the first time by Fontana in 1781 “corps oviforme”, and then described by Valentin in 1836 [1]. These early microscopists set up the foundations to study the most exciting organelle inside the nucleus: the nucleolus. Later on, Barbara McClintock first proposed, in *Zea mays*, that nucleoli assemble around the Nucleolar Organizer Regions (NORs), which happened to be the coding sequences for the ribosomal RNAs (rRNA) [2]. These special rRNA genes are in multiple copies per genome since rRNAs are required in large amounts and they cannot be amplified like proteins during translation. Thus, cells rely on having multiple rRNA gene copies that are transcribed very actively to fulfil cell demands. These copies are often grouped in tandem arrays traditionally referred to as the ribosomal DNA array (rDNA). In addition to the rDNA, the nucleolus comprises pre-ribosomal particles and pre-rRNA with assembly factors that will progressively mature in the nucleolus, then in the nucleoplasm and finally into the cytoplasm [3]. For many decades, the nucleolus was considered a mere rRNA and ribosome factory, and only recently the myriad of processes that it is involved in have begun to be unraveled [4,5,6]. Furthermore, the nucleolar structure has proven as an excellent maker for cell stress and growth status. Indeed, nutrition status, environmental cues and stress impinge on its size and shape [7,8]. The nucleolus is also the assembly site and processing for small nuclear RNAs (snRNAs), small nucleolar RNAs (snoRNAs), telomerase and the signal recognition particle (SRP); all of them with physiological implication for the cell that go beyond the nucleolar environment [9]. To get a much deeper knowledge on the nucleolus in different organisms, we refer the readers to previous excellent reviews by [10,11]. In this review, we will concentrate on the structural changes of the nucleolus (and the rDNA) of the budding yeast *Saccharomyces cerevisiae*. However, we will briefly describe some aspects of the nucleolar biology of metazoans that are connected to nucleolar biology of this simple yeast. This correlation has boosted *S. cerevisiae* as an excellent model organism whereby we can better understand how stress influences the nucleolar biology and vice versa. We will mainly focus on how different stresses modify the morphology of the nucleolus. Importantly, we are mostly leaving out of this review the long-term effects that clastogenic and aneugenic stresses can cause on the rDNA structure, which can also have a great and sustained impact on the nucleolar morphology. The connections between these stresses, the rDNA/nucleolus, cancer and aging deserve by itself a deep and updated review in the future. In summary, we have organized this review in the following chapters. First, a brief historical overview on the nucleolus and its usefulness as a marker of cancer and aging, followed by a description of its composition and structure in the budding yeast. Next, we will go over the morphological changes in the nucleolus seen in both a normal unperturbed cell cycle and after stress. We will pay attention to what is known about the underlying mechanisms responsible for nucleolar reorganization and why we now know they intimately connect with aging. Finally, we will briefly review a clear case of a connection between nucleolar stress and control of cell proliferation in tumor cells that might be exploited in antitumoral therapy. Before we proceed into the next section, we introduce some concepts to better understand the processes presented here. According to Roger B. McDonald’s *Biology of Aging*, aging is the random change in the structure and function of molecules, cells, and organisms that is caused by the passage of time and by one’s interaction with the environment. Aging increases the probability of death. On the other hand, longevity is defined as the potential maximum age that an individual of a particular species can attain, that is, the evolved length or duration of life for a species; while life span is the length of life of an individual cell, organ or organism [12]. In budding yeast, aging has been traditionally divided into two different concepts. The first concept, replicative aging, stems from early works in the 50s, and is defined as the number of divisions an individual yeast cell undergoes before dying. The second concept, chronological aging, is the length of time a population of yeast cells remains viable in a nondividing state following nutrient deprivation [13,14]. Both terms are also usually referred as replicative life span and chronological life span based on the assays developed to measure them [15]. On the next section, we will explain very briefly the evidences for the nucleolar size in relation to cancer and aging.

## 2. The Nucleolus as a Marker of Cancer and Aging in Metazoans

The size and shape of the nucleolus are tightly related to the growth and proliferation status of the eukaryotic cell, and this is known since several decades ago [16,17]. An increase in size correlates with a high growth and proliferation rate [18], and this feature has been used by pathologists over decades as a prognostic marker in tumour tissue samples [19]. In fact, for a specific type of cancer, the Clear Cell Renal Cell Carcinoma (ccRCC), the nucleolus is used as the main marker to establish the histological grade [20,21]. Tumour samples with a normal nucleolar number, size and shape correlates with a good prognostic, while an enlarged size and/or irregular shape correlates with a bad prognostic [22,23,24,25]. Ribosome biogenesis is thus becoming an important aspect of cancer biology [26,27,28]. Remarkably, nucleolar size and aging are also correlated; small nucleoli and reduced ribosome biogenesis are hallmarks of longevity, whereas expanded nucleoli, elevated ribosome biogenesis and increased protein synthesis are hallmarks of aging [29]. On the one hand, long-lived mutants of *Caenorhabditis elegans* (e.g., daf-2), *Drosophila melanogaster* (under dietary restriction, rapamycin treatment or insulin-like mutants), mice (under dietary restriction and IRS1 long-lived mutants) and human muscles biopsies (under modest dietary restriction) showed the small nucleoli phenotype [30]. On the other hand, cells from Hutchinson-Gilford progeria syndrome (HGPS) patients and cells from aged healthy individuals showed the expanded nucleoli phenotype [31]. In yeast, nucleolar structure goes through two different phases during aging as well: first, expansion and, secondary to this, fragmentation [32,33]. The above observations, conserved throughout evolution, could be employed as a predictive cellular marker for aging in both healthy and aged individuals [34,35].

## 3. The Structure of the Nucleolus and the rDNA in the Yeast *Saccharomyces cerevisiae*

Nucleolar structure can be dissected in simpler model organisms, like the budding yeast *Saccharomyces cerevisiae*. An extensive knowledge about nucleolar biology does exist in this organism, ranging from rDNA structure and stability to rRNA transcription, processing and ribosome assembly. Detailed reviews about these aspects of the nucleolar and rDNA biology can be found elsewhere and are out of the scope of this review [36,37,38]. However, we will briefly summarize several facets about the nucleolar structure to consider for the understanding of how stress influences yeast nucleolar biology. In budding yeast, the nucleolus is a crescent-shaped structure that occupies roughly one-third of the nuclear volume. The rDNA is enclosed inside it, abutting the nuclear envelope. The rDNA is located on the right arm of chromosome XII, the largest chromosome in this yeast species. The basic 9.1 Kb unit of the rDNA is repeated in a tandem head-to-tail continuous array between 100 and 200 times per locus, half of them being transcriptionally active [39]. Besides the transcription units, two non-transcribed spacers (NTS1 and NTS2), also known as intergenic spacers (IGS1 and IGS2), lie between the 5S transcription unit and the 35S. The 35S is transcribed by the RNA polymerase I (RNApol I) and forms the precursor rRNA for the 25S, 5.8S and 18S rRNAs, while the 5S is transcribed by RNA polymerase III (RNApol III). A bidirectional criptic non-coding promoter (E-pro) in the IGS1 region is transcribed by the RNA polymerase II (RNApol II) and is subject to silencing by the sirtuin Sir2, an NAD+-dependent histone deacetylase. Two other important elements are an origin of replication (ARS) in the IGS2 region and a replication fork barrier (RFB) in the IGS1 region. The protein Fob1 at the RFB avoids collision between the transcription and the replication machineries (Figure 1).

Fob1 is also necessary to induce double strand breaks (DSB) and homologous recombination (HR) at this site, a mechanism used for expansion and contraction of the rDNA [40]. Sir2 regulates the recombination between different copies of the rDNA [41], tuning a sister chromatid equal/unequal HR pathway that depends on the E-pro transcriptional status [42]. An alternative non-HR pathway is also involved in the amplification of the rDNA array [43]. Different pathway choice of amplification was proven to be modulated by nutrient availability, and this, in turn, requires TOR signalling over multiple histone deacetylases (HDACs) of the sirtuin family [44]. Very recently, a model has been proposed whereby Sir2 is regulated by the amount of upstream activator factors (UAF), thus affecting the rDNA recombination outcome (amplification or maintenance). This model represents a mechanism to count and adjust the rDNA copy number [45]. One important feature of old yeast cells (and of *sgs1* and *sir2* mutants), is the formation of extrachromosomal rDNA circles (ERCs); these can cause aging, presumably by their accumulation leading to nucleolar enlargement and fragmentation [46]. The rDNA is subject to perinuclear membrane attachment through the inner nuclear membrane (INM) chromosome linkage INM proteins (CLIP) and mitotic monopolin complex (Cohibin) [47]. CLIP (Heh1 and Nur1 in yeast) and Cohibin (Csm1 and Lrs4) are also involved in rDNA silencing and stability through tethering of the rDNA [48]. The rDNA is tightly associated to this perinuclear membrane [49] in order to keep it aside from the HR machinery [50]; the rDNA is the most unstable region in the genome due to its repetitive nature and high recombination rate [51]. Interestingly, the nuclear envelope adjacent to the nucleolus was shown to have different properties and abilities during membrane expansion [52]. Separation of the nucleolus from the rest of the genome is thought to emerge through differential physical properties [53,54], resulting in different aggregation and phase separation, either as a polymer or as a liquid phase [55,56]. Although not entirely proven, rDNA size, nuclear envelope metabolism and liquid phase properties of the nucleolus contribute altogether to its actual shape and morphology. In addition, rDNA condensation seems to play a central role in quickly reshaping the nucleolus within a cell cycle, as we describe in the next chapter.

### Morphological Changes of the Yeast Nucleolus during the Cell Cycle

During a single cell cycle, the copy number of the rDNA array is thought to change little. Nevertheless, its morphology under the microscope goes through astonishing changes. Pioneering works using fluorescence in situ hybridization (FISH) proved that the rDNA in G1 is organized in locally-constrained disseminated clusters i.e., the rDNA units are stained by the FISH probe as scattered foci within the nucleolar space [57]. This structure was referred to as puff and likely represents clusters of condensed rDNA units connected by strings formed by other less-condensed units. When cells reach G2/M, the rDNA is either clustered in a single focus or is forming an arc at the nuclear periphery [58]. Protracted G2/M arrest make the rDNA to protrude out of the nuclear mass and adopt the most spectacular structural reorganization observed for any yeast chromosome region: the rDNA loop [57,59,60]. This loop is not only seen by FISH but with other in vivo techniques that used GFP chimeras for proteins that tightly bind to the rDNA units; e.g., Fob1 and Net1 [61], or LacI-GFP bound to lacOs inserted all along the rDNA [62]. Much more recently, similar observations on the rDNA structure have also been obtained by a modified FISH-DAPI technique [63] and through the novel CRISPR/dCas9-GFP imaging technology [64]. The rDNA loop has been extensively studied as a model for the chromosome condensation process that occurs in metazoan prophase, despite the controversy about whether yeast chromosomes actually condense as their higher eukaryotes counterparts. Regardless, the amenability of yeast genetics was determinant to test how the structural maintenance of chromosome (SMC) complexes shape mitotic chromosomes [58,59,65,66,67,68,69]. In these works, it was firmly established that both condensin and cohesin SMC complexes were essential for the transition from puff to loop. Another common partner with SMC complexes and key player in the condensation of metazoan chromosomes, topoisomerase II (Top2), proved to be dispensable [59]. Not only SMC complexes and Top2 have been assessed for their role in the rDNA structure, several cell cycle regulators play critical roles in rDNA reshaping. This is especially important after anaphase onset, where the loop is lost but rDNA condenses back into short lines or clusters [58,61]. Among cell cycle regulators that control rDNA morphology at this stage we find the Polo Like Kinase 1 ortholog Cdc5, Aurora Kinase B ortholog Ipl1 and the master cell cycle phosphatase Cdc14 [58,61,70,71,72].

## 4. Key Facts about Yeast rDNA Transcription and Ribosome Production

The fine structure of the rDNA was established by electron microscopy (Miller spread technique) [73,74,75], and the nucleolar ultrastructure localization, assembly and organization by mutant analysis [76,77]. The arrangement, in the case for the Miller spread, was similar to that of a ‘Christmas tree’, the stem or trunk formed by the rDNA, while ongoing transcription (rRNA) forming the branches, with knobs (representing Christmas baubles) being the rRNA processing machinery (SSU processome). Non-transcribed units appeared as ‘naked’ rDNA without branches [78,79]. Moreover, transcription of the rDNA by RNApol I and rRNA processing and modification take place temporally and spatially in a concomitant manner [80]. Three striking characteristics of the nucleolus highlight its central role in cell growth and metabolism, as well as the enormous resources it consumes. Firstly, RNA polymerase I (RNApol I) transcription activity (60 nt/sec) accounts for 60% of the total transcription in growing yeast, and 50% of total RNA polymerase II (RNApol II) activity is devoted to ribosomal proteins (RPs) genes. Secondly, the high rate of ribosome production (approximately 2000 per min) accounts for 50% of the total protein mass in the cell. Thirdly, most of the splicing resources (90%) are invested in messenger RNA (mRNA) coding for RPs [81,82]. This clearly indicates that transcription of this locus and ribosome biogenesis must be tightly coupled and that nutrient status and availability, as well as environmental conditions (internal and external), have a profound effect on the nucleolus [44,83,84,85]. In fact, ribosome biogenesis and cell size are related through Sfp1 transcriptional regulation [86,87,88], as well as to the S6 kinase (Sch9 in budding yeast and Sck2 in fission yeast) [89]; both regulated, in turn, by the nutrient sensing TORC1 complex [90,91,92]. Moreover, cell growth and cell division are coupled through ribosome content, translation initiation rate and the G1 cyclin *CLN3* expression [93].

## 5. TORC1, Stress and the Size of the Nucleolus

The Target of Rapamycin (TOR), master regulator of the cell, coordinates metabolism, cell growth and proliferation. TOR is composed by the TOR complex 1 (TORC1) and TOR complex 2 (TORC2) in budding yeast, mTORC1 and mTORC2 in humans. The TORC1 complex is rapamycin sensitive and mainly involved in cell growth by promoting ribosome biogenesis, translation initiation, metabolism and cell cycle progression, while inhibiting, on the other hand, stress responses and autophagy [94,95]. TORC2 complex is rapamycin insensitive and is involved in actin polarization, cell wall integrity, endocytosis and sphingolipid biosynthesis [96,97]. Two TOR genes are found in *S. cerevisiae* (*TOR1* and *TOR2*) as opposed to only one TOR gene in humans [98]. Tor1 only binds to TORC1, while Tor2 (present also in TORC2) binds very weakly and transiently to TORC1 in *tor1*Δ cells and cannot compensate for the absence of Tor1 in this complex [99,100,101].The TORC1 complex controls all three RNA polymerases [102,103]. In particular transcription of the 35S rDNA by RNApol I, transcription of RP and Ribosomal biogenesis genes (Ribi) by RNApol II [104,105], and transcription of the 5S rDNA and transfer RNA (tRNA) genes by RNApol III [106]; all needed in order to build ribosomes and synthesize proteins [107,108,109]. Remarkably, Tor1 binds to the 35S promoter of rDNA in a rapamycin and nutrient starvation manner [110], although the exact mechanism whereby Tor1 controls the rDNA transcription is still not clear [111,112,113], nor whether a nuclear resident complete TORC1 pool exists [102,114]. In addition to 35S promoter binding, Tor1 binds to the 5S promoter, upregulating transcription by the RNApol III [115], which in turn limits lifespan [116]. Besides, TORC1 controls translation initiation and protein synthesis through phosphorylation of Sch9 (S6 Kinase in humans), and through the stability of eIF-4G and phosphorylation of eIF-4EBP1, both of them eukaryotic initiation factors [91,117,118,119,120,121]. In other terms, protein translation accuracy has been linked to proteostasis and lifespan modulation in *C. elegans* [122]. Likewise, translation inhibition and decreased protein metabolism is linked to lifespan extension and improved proteostasis in this nematode, as well as in mammalian cells [123,124,125]. Both lifespan extension and improved proteostasis are influenced by dietary restriction and TOR signalling [126,127,128], establishing TORC1 as a sensor of proteostasis [129,130,131,132]. Certainly, the rDNA/nucleolar structure is influenced by inhibitors of the TORC1 complex [133,134,135]. A dramatic shrinkage or compaction of the rDNA, both in cycling and mitotic arrested cells, takes place upon exposure to different stressor such as rapamycin, nitrogen starvation, glucose starvation, calorie restriction, heat stress (HS), oxidative stress and UV radiation, as well as by genetic manipulation of the TORC1 function [63,64,133,135,136] (Figure 2).

Former stressors, with the exception of UV, are well-known inhibitors of the TORC1 complex [91,137,138], with an outcome (for some of them) of cease in new ribosome production [139]. Nonetheless, depletion of 60S ribosomes subunits leads to yeast life span extension [140]. Although there is no agreement on the terminology used for this phenomenon (condensation, hypercondensation, compaction, contraction or clustering), all these stresses certainly cause a shrinkage and dramatic reorganization of the nucleolus into a smaller and more compact structure; a small nucleoli phenotype. A model summarising the above interpretations is presented (Figure 3). One last interesting observation refers to the involvement of the INM proteins (CLIP), monopolin (Cohibin) and the nuclear envelope in rDNA and nucleolar repositioning under TORC1 inhibition. It was found that part of the nucleolar proteins became separated from the rDNA, and subjected to autophagy, while the rDNA was preserved from this [141]. Whether or not this is related to nucleophagy induced by caloric restriction is a question to explore in the future [142].

### TORC1, Condensin, Epigenetics and the rDNA Chromatin Compaction

The TOR pathway is involved in the rDNA chromatin structure in different ways; epigenetic regulation and condensin activity appear as potential mechanisms. For example, one mechanism might operate through the regulation of the localization of RNA pol I and the Rpd3 histone deacetylase to rDNA [133], which leads to histone H4 deacetylation and, in turn, to condensin loading onto the array [143]. Upon nutrient availability, TOR inhibits Rpd3 [144], but this is relieved under starvation or rapamycin treatment, leading to condensin accumulation at the array. In this scenario, condensin acts on the stability of the rDNA [145,146]; hence, condensin-mediated nucleolar shrinkage could prevent genetic instability associated with repetitive sequences. On the other hand, Hmo1, a High Mobility Group protein (HMG) involved in the specialized chromatin state at the rDNA [147,148], is regulated by TORC1 [105,149,150]; and Hmo1 was required alongside with condensin to mediate both a starvation-induced transcriptional position effect within the rDNA and nucleolar contraction [151]. Mitotic rDNA condensation is also regulated by the Jhd2 demethylase, which acts on histone H3, the maintenance of Csm1/Lrs4 (Cohibin) and condensin association with the rDNA [152]. How is condensin activated outside anaphase under stress conditions? It was shown that the condensation (prior to anaphase) of this locus under stress conditions was independent from Cdc14 activation [135,153], the phosphatase required for rRNA transcription inhibition and condensin loading onto the rDNA during anaphase [154,155,156]. One possible explanation is through the atypical Rio1 kinase, involved in both ribosome biogenesis and nutrient sensing parallel to TORC1 [157,158]. Noteworthy, segregation of the rDNA during anaphase is also supported by Rio1, leading to downregulation of RNApol I, stimulation of rRNA processing, rDNA condensation and Sir2 recruitment to the rDNA [159]. Moreover, it was suggested that condensation could occur differentially between active and inactive rDNA repeats by partial downregulation of RNA pol I [160]. Possible mechanisms include condensin overloading, post-translation modifications of condensin and/or epigenetic mechanisms. Regarding the latter, histone modifications (e.g., acetylation), sirtuins and other epigenetic regulators are possible candidates [161]. The sirtuin proteins Hst3 and Hst4, have been shown to be modulated by the TORC1 complex, affecting the acetylation status of histones H3 and H4, independently of the nicotinamidase *PNC1* gene expression [162]. It is important to stress that sirtuins are regulated by NAD+ levels, which depend on two pathways: *de novo* pathway (from tryptophan) and the NAD+ salvage pathway, the latter governed through the Pnc1 nicotinamidase [163,164]. Moreover, other members of the sirtuin family such as Sir2 and Hst2 are part of the same longevity pathway involving TOR and calorie restriction. This pathway operates through the upregulation of the PNC1 gene, leading to higher NAD+ levels, and hence, increasing sirtuin activity and promoting longevity through the stabilization of the rDNA [164]. In this respect, it was shown that TORC1 inhibition leads to histone deacetylation, and Sir2 enhanced association to the rDNA, through Pnc1 and Net1 [134]. In addition, silencing of the rDNA array by Sir2 is regulated through condensin action, and rapamycin treatment increases condensin binding to the rDNA [165,166]. Axial contraction and short-range chromatin compaction were shown to rely partially on the deacetylase Hst2 (possibly acting on condensin) in order to promote chromosome condensation in anaphase by removal of acetyl groups from histone H4 [167,168]. Thus, it is indeed feasible that sirtuins may directly control rDNA condensation. Post-translational modifications of histones are considered to promote chromatin compaction [169]. Remarkably, H4K16 deacetylation is important for chromatin compaction by promoting H2A and H4 interaction [170,171], and this epigenetic mark at subtelomeric regions also regulates yeast lifespan [172]. In addition, modifications mediated by acetyl transferases like Nat4, and the loss of histone H4 acetylation, were linked to calorie restriction-mediated longevity and to rDNA silencing [173,174]. Besides, chromatin remodelers like INO80, histone deacetylases like Rpd3 (mentioned above), and histone chaperones like the FACT complex are needed in order to modulate TORC1 signalling onto chromatin and to facilitate ribosomal DNA nucleosome assembly and transcription [175,176,177]. As an example, Histone H3 acetylation at lysine 56 is regulated by TORC1, enabling rDNA transcription and nascent rRNA processing [178]. Finally, in the case of the FACT complex, a small nucleolar phenotype (conserved through evolution) was found in mutants for the Spt16 and Pob3 subunits of this complex [179]. Whether all these modifications have an impact on the nucleolar structure is a question for future research. What is clear is that the rDNA is subjected to epigenetic silencing [180,181,182,183]. Sir2-dependent and independent (as well for other members of the sirtuin family: Hst2, Hst3, Hst4) life span extension mechanisms are a topic of hot debate [164,184,185,186,187,188,189,190,191,192]. Even though sirtuins appear as conserved regulators of aging/longevity [193], their inclusion in a TOR mediated nucleolar compaction and lifespan extension model needs further clarification [194,195].

## 6. Nucleolar Stress Remodelling and p53 Stabilization in Cancer

As in *Saccharomyces cerevisiae*, different nucleolar rearrangements do occur in higher eukaryotes during stress, the cell cycle or as a result of cell aging. These morphological changes take place despite obvious differences in nuclear and nucleolar physiology; namely, a closed mitosis without nucleolar disassembly in the case of the budding yeast versus an open mitosis with transient nucleolar disassembly in higher eukaryotes [10]. Another major difference lays on the different architectural complexity of the nucleolus. In higher eukaryotes a define set of compartments can be easily distinguished under the microscope: the fibrillary center (FC), the dense fibrillary component (DFC) and the granular component (GC), i.e., a tripartite compartmentalization. By contrast, in the budding yeast only a bipartite compartmentalization, comprising fibrillar and granular components, is present. The main differences are summarised in Table 1.

Each compartment undertakes a specific task; traditionally, FC is where rDNA transcription occurs, DFC where rRNA processing takes place and GC is dedicated to late rRNA processing and pre-ribosomal particles assembly [11]. Thus, the alteration and reorganization of these spatial structures are indicative of alteration in nucleolar function [196]. Remarkably, nucleolar morphology, visualised by NORs silver staining (AgNORs), is correlated to tumour DNA content and ploidy; aneuploid tumours have higher AgNORs counts [197,198]. This could be due to amplification of the five human acrocentric chromosomes (13, 14, 15, 21 and 22), where the NORs are located, as proposed in [23]. On the other hand, rDNA copy number variation, increasing the copies of 5S genes while decreasing the 45S copies, have been linked to nucleolar activity, proliferation and inactivation of p53 [199]. All these rDNA changes may operate as a tumour adaptive strategy to promote genome instability.

Nucleolar stress presents here four main features, a decrease in the size and volume of the nucleolus, redistribution and/or fragmentation of the nucleolus, inhibition of rRNA production by RNApol I, and relocalization of nucleolar proteins into the nucleoplasm [200,201]. Different stresses that impact on the nucleolar structure can have different outcomes. On the one hand, there are stresses that produce nucleolar fragmentation/disintegration; whereas, on the other hand, there are other stresses that alter nucleolar size but not its integrity [7,8]. Among the former, we find classic antineoplastic drugs like mitomycin C, cisplatin and oxaliplatin, mitoxantrone, doxorubicin, camptothecin (topotecan and irinotecan) and actinomycin D; all but the latter being compounds that generate DNA damage. Specifically, nucleolar aggregates or caps are formed upon DNA damage and actinomycin D treatment [202,203]. In this respect, it has been suggested a specific DNA damage response (DDR) for the nucleolus (n-DDR), whereby distinct repair features promote the integrity of the rDNA [204]. By contrast, inhibitors of the late rRNA processing such as 5-fluorouracil and homoharringtonine, and TOR inhibitors like rapamycin and its derivatives (temsirolimus and everolimus), belong to the second category of stressors [205,206]. New compounds such as CX-5461 and CX-3543, selective inhibitors of rDNA transcription by the RNA pol I, show nucleolar disintegration and nucleolin redistribution, respectively [207,208,209]. Whereas stressors that fragment the nucleolus could be genotoxic, those that suppress rDNA transcription, rRNA processing and ribosome production could be safer antiproliferative therapies. Other potential stress treatments with such profile, alone or in combination with current antitumor chemotherapy and radiotherapy, could be hypethermia and calorie restriction [210,211,212,213]. Finally, a compound with antimetastatic potential described recently, called metaarrestin, acts through the inhibition of transcription by RNA pol I, reducing the nucleolar volume [214]. Interestingly, stresses with a biological origin, like viral infections, also lead to nucleolar alteration (e.g., enlarged FC) [215,216,217]. Another interesting phenotype, the formation of nucleolar aggresomes, is related to the wrong nucleolar turnover of p53 in aging and progeria [218], as well as to proteotoxic stress, serving the nucleolus as a hub for misfolded proteins storage and proteostasis control. This has been recently reviewed in the context of liquid-liquid phase separation and liquid-solid phase transition of the nucleolus and their role in cancer and neurodegenerative diseases [219]. Finally, we will briefly discuss p53 function in the nucleolus, even though other reviews in this special issue are covering different aspects of it. p53 is a protein regulated at different levels, and its activation upon nucleolar stress depends on the p53-Mdm2 axis. The binding of p53-Mdm2 renders p53 inactive under non-stressed conditions. This happens through the ubiquitin ligase activity of Mdm2 and subsequent p53 degradation by the proteasome. We must mention that 60% of tumours have mutant *TP53* [220], yet, regardless of this p53 mutant status, there are several p53 isoforms that have an impact on p53 transcriptional activity and on tumour progression [221,222,223,224]. It would be valuable to see whether there is a connection between these isoforms and the p53-Mdm2 axis. In response to stressful conditions, several ribosomal proteins are released from the nucleolus into the nucleoplasm: RPL11, RPL23, RPL5 and RPL7. There, they bind to Mdm2, which inhibits the destruction of p53. Moreover, another RP protein, RPL26, binds to the 5’-UTR of p53 mRNA, enhancing its transcription under DNA damage [225]. This nucleolar stress mechanism indicates the cell the synthesis and ensemble status for rRNAs and RPs, establishing a quality control surveillance mechanism [226]. One of the proteins involved in this sensing mechanism is PICT1/GLTSCR2, the homologue of the yeast ribosome biogenesis factor Nop53. PICT1 is bound to RLP11, avoiding its release into the nucleoplasm and hence the binding to Mdm2. This makes Mdm2 available for p53 binding as mentioned above [227]. Besides this, PICT1 also stabilises the tumour suppressor PTEN [228]. When PICT1 is absent from the nucleolus (*Pict1*-/- or low levels of PICT1), RPL11 is released to inhibit Mdm2. Although PICT1 may function differentially, as a tumour suppressor or as an oncogene, depending on the environment and conditions, low levels of PICT1 have been found on ccRCC with an inverse correlation to the Fuhrman grade system, which classifies tumours based on nuclear/nucleolar abnormalities [229]. In breast cancer tumours, low levels of PICT1 are associated to tumour progression [230], while cytoplasmic expression of this protein are related to a bad prognostic for non-small cell lung cancer [231]. Finally, PICT1 suppression under hypoxic conditions in glioblastoma tumour cells augments the survival and invasiveness of the tumour [232]. It is possible that tumours cells, subjected to endogenous or exogenous stress, could modulate PICT1 levels as a strategy to impinge on p53 function and/or promote their growth by ribosome biogenesis, depending on the needs. This enhances this nucleolar axis as a putative target in antitumour therapy.

## 7. Future Perspectives in Nucleolar Stress and Health: From Yeast to Humans

Life-span extension can be achieved by calorie restriction (glucose, protein, amino acids deprivation), or by pharmacological interventions, such as rapamycin and other small molecules (e.g., metformin and resveratrol) [233,234,235]. Side effects from acute or long-term treatments could be diminished by intermittent and/or very low doses of rapamycin treatment, fasting-mimicking diets, intermittent fasting, among other regimes [236,237,238]. The exact extent of stress (quality, quantity and intensity) and how it affects different cell types, developmental stages and disorders, needs to be evaluated in order to poise the cell and the organism into a protective mode. Although we have focused on a TOR centred view in yeast, other signalling routes, interconnected with TOR itself, like Growth Hormone (GH), Insulin/IGF-1-like, and PI3K-AKT are however important players in more complex higher eukaryotes. Other age-related processes such as mitochondrial dysfunction, also influenced by sirtuins and TORC1, should be taken into account as well [193,239,240]. Inhibiting TOR signalling could mediate longevity through both sirtuin and ribosome production (amid others) and its effect on (but not only) rDNA and nucleolar structure and stability. Thus, the nucleolus could be serving as both a genome buffering system and a stress sensor for the cell [241]. In fact, a whole rDNA theory of aging in budding yeast has been proposed by Kobayashi [242]. In addition, a role for the rDNA as an evolutionary conserved clock of aging has been postulated as well [243]. Could a central hub or integrator in the cell, such as TOR, be communicating (through its inhibition) the stress signals into a sensor or buffering system like the rDNA/nucleolus, linking growth and stress and, in that way, enhancing stress responses and protecting the rest of the genome from further damage? This is an interesting hypothesis that remains to be further elucidated. In relation to cancer biology, the use of the nucleolus as a marker in a wide range of tumours continues as a valuable prognostic tool for the pathologist. Meanwhile, the specialized nucleolar nature opens new avenues for potential treatments with less genotoxicity. Finally, one clear conclusion is drawn, the nucleolus is dramatically reduced upon TOR inhibition or caloric restriction, and this is related to longevity. By contrast, an enlarged nucleolus is indicative of aging. This should be used as a predictive hallmark (among others) in aging and longevity research with potential translation into the clinic [244].

## Figures and Tables

**Figure 1 cells-08-00779-f001:**
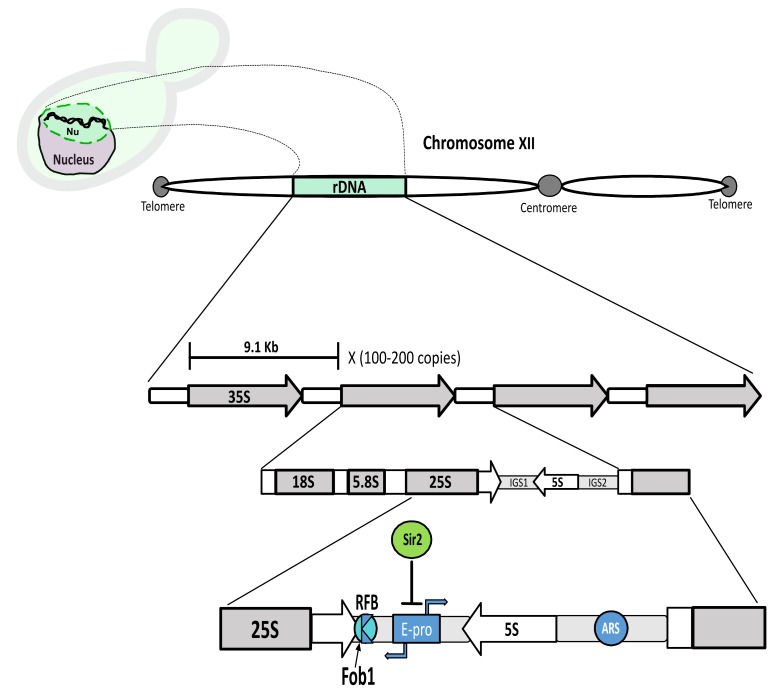
Schematic representation of the ribosomal DNA array; **Top lef:** A yeast cell with the rDNA portrayed as a two coiled chains (black) inside the nucleolus (**Nu**) (green), which occupies the upper part of the nucleus (light purple) in this drawing. **Top middle:** The rDNA (green) is located on the right arm (here left) of chormosome XII. **Middle:** Representation of the basic 9.1 Kb unit, repeated 100–200 times in tandem. The 35S transcription unit (transcribed by the RNApol I) is depicted (18S, 5.8S and 25S). These are separated by internal transcribed spacers (ITS1 and ITS2) (not shown), besides external transcribed spacers which lie at the 18S and 25S ends (not shown). The 35S and the 5S are separated by two intergenic regions (IGS1 and IGS2). **Bottom middle:** Specific features in the IGS1 and IGS2 regions. IGS1: E-pro, cryptic bidirectional promoter (RNApol II), silenced by Sir2; RFB, replication fork block. Binding of Fob1 at RFB, creates a unidirectional barrier for oncoming replication to avoid collision with ongoing transcription from the 35S. Direction of arrows represents direction of transcription; IGS2: ARS, origin of rDNA replication.

**Figure 2 cells-08-00779-f002:**
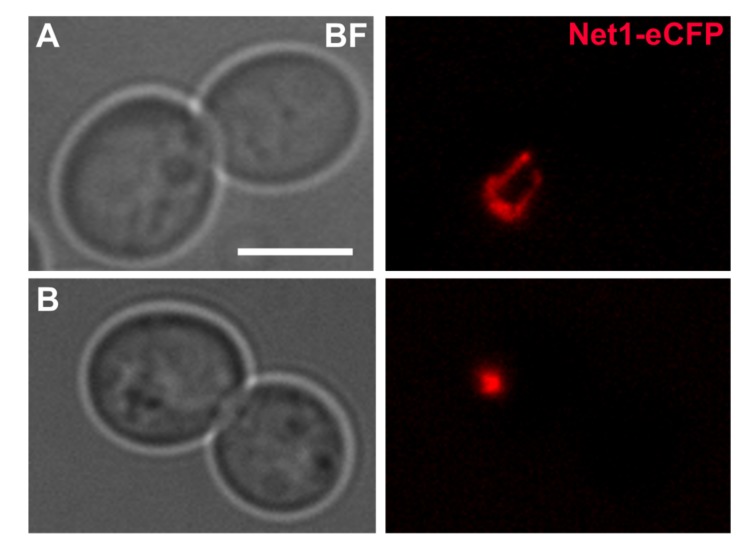
The effect of stress in the nucleolus of yeast cells blocked at G2/M. (**A**) Typical dumbbell morphology of a nocodazole (Nz) G2/M arrest at 25 ${}^{\circ }$C with the rDNA metaphase loop; the strain carries the Net1 protein with an eCFP tag (pseudo coloured red). Bar, 5 μm. (**B**) A cell culture coming from a Nz arrest was stressed by shifting the temperature from 25 ${}^{\circ }$C to 37 ${}^{\circ }$C for 90 min, while keeping the Nz arrest. A dramatic compaction of the rDNA signal is observed upon HS. Strain background: W303. This is original work.

**Figure 3 cells-08-00779-f003:**
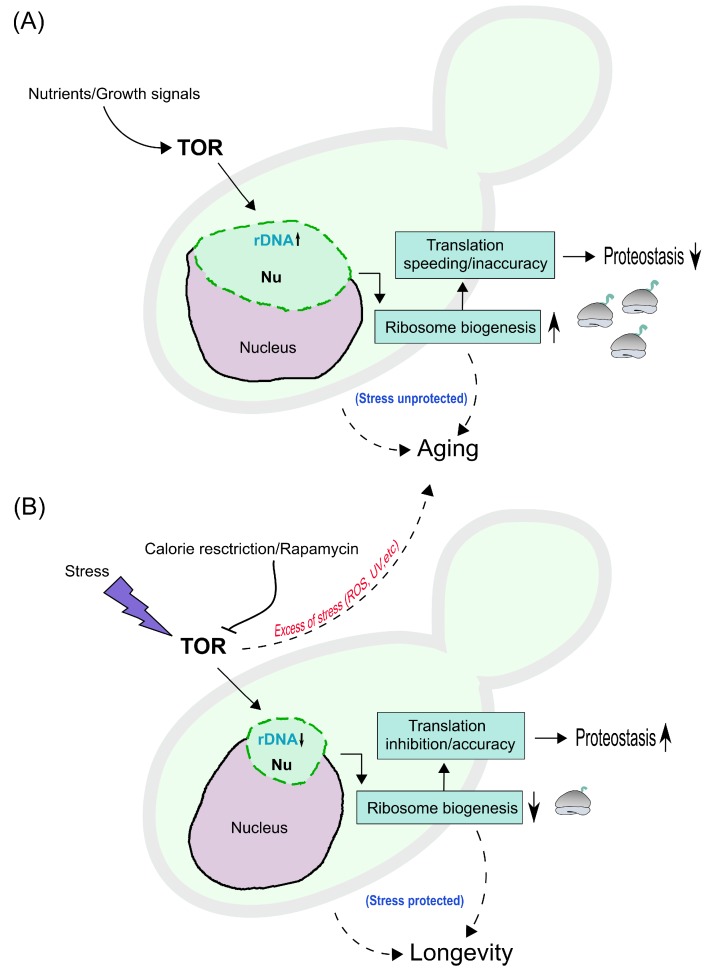
Hypothetical model for nucleolar condensation and aging/longevity mechanisms in yeast cells. (**A**) Nutrients and growth signals stimulate the Target Of Rapamycin (**TOR**), leading to rDNA transcription in the nucleolus (**Nu**), which acquires an enlarged nucleolar phenotype (green); as a result, ribosome biogenesis increases, depicted on the right as ribosomes (grey) with ongoing translation (green ribbons). This impinges on the rate and accuracy of translation, increasing the former while decreasing the latter, thus compromising proteostasis. This will lead in turn to an ’unprotected stress’ mode and to aging (dashed line with arrow on the right). Other intrinsic mechanisms dowstream of TORC1 (e.g., mitochondrial dysfunction, autophagy inhibition, etc.) can additionally lead to aging (dashed line on the left). (**B**) Calorie restriction, rapamycin and stress inhibit TOR, leading to rDNA transcription inhibition and to condensation of the nucleolus (small nucleolar phenotype); reducing ribosome biogenesis, while inhibiting translation rate and/or increasing accuracy of translation, with improved proteostasis. This will lead in turn to a ’protected stress’ mode and to lifespan extension and longevity (dashed line on the right). Other intrinsic mechanisms can additionally lead to lifespan extension (dashed line on the left). Nevertheless, excess of stress (ROS: Reactive Oxygen Species, UV: Ultra Violet light, etc) can lead to aging (dashed line pointing towards panel A).

**Table 1 cells-08-00779-t001:** Structural differences between the yeast and human nucleolus and rDNA. RENT: regulator of nucleolar silencing and telophase exit; NoRC: nucleolar remodeling complex; eNoSC: energy-dependent nucleolar silencing complex.

Nucleolar and rDNA Features	
Yeast	Humans
Nucleolus and nucleus are not disassembled in mitosis	Nucleolus is disassembled in mitosis along with the nucleus
Bipartite composition	Tripartite composition
Absence of a perinucleolar domain	Presence of a perinucleolar domain
All rRNA genes together	5S and 45S genes in different genome loci
The rDNA in a single array at chromosome XII right arm	The rDNA 45S array in the five acrocentric chromosomes (13, 14, 15, 21, 22); 5S in chromosome 1
100–200 copies (haploid) of a 9.1 Kb unit	300–400 copies (haploid) of the 43 Kb unit (45S)
The rDNA is attached to the nuclear envelope	The rDNA is not always attached to the nuclear envelope
Presence of a cryptic RNApol II promoter at the rDNA	Absence of a cryptic RNApol II promoter at the rDNA
Silencing complexes: RENT (Net1, Sir2, Cdc14); Tof2-Lrs4/Csm1	Silencing complexes NoRC (TIP5, SNF2h); eNoSC (SIRT1, NML, SUV39H1)

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
