# Peer review of "Nucleolar and Ribosomal DNA Structure under Stress: Yeast Lessons for Aging and Cancer"

_cells, 2019, doi:10.3390/cells8080779_

Round 1

Reviewer 1 Report

Manuscript ID: cells-544996

Type of manuscript: Review
Title: Nucleolar and Ribosomal DNA Structure Under Stress: Yeast Lessons for
Aging and Cancer
Journal: Cells

This review by Matos-Perdomo and Machin summarizes the functions played by the nucleolus in the context of aging and cancer, in particular based on work performed in yeast. The topic is extremely interesting and novel. The manuscript is very well-written and of broad interest.

The Authors provide important information about the various and complex level of alterations nucleolar function and integrity undergo, and this is a crucial (and yet mostly overlooked) aspect for the understanding of the functional consequences of nucleolar activity in physiological and pathophysiological systems.

I have a few concerns that will be important to address before publication to further improve the manuscript understanding for broader readerships:

1)   It is somehow difficult to follow the sequence of the different chapters. On page 2 line 39, it would be convenient to provide the summary of the way the review is structured.

2)   It would be useful to clearly define from the beginning what is meant by longevity and what by ageing in yeast and in general in complex organisms.

3)   The Authors cite 229 references and this is certainly an immense effort. Nevertheless, in some cases it would be useful to describe the content a bit more in details, for example on page 4 related to refs from 48 to 52. The text is too compact and the content results difficult to follow.

4)   The data shown in Figure 2 require a reference. Is this original work?

5)   There is an imbalance between the aging and cancer part and a gap between the cancer part and the studies in yeasts. Also in the abstract it should be more specifically expressed what aspects of cancer biology are related to nucleolar stress response. Certainly p53 is a major link but it should be explained in more detail, for example by summarizing similarities and differences between yeast and higher eukaryotes in a table. Moreover how the work in yeast was helpful for cancer biology should be clarified more explicitly.

Minor points:

-       abstract line 10 should be NSR and not NRS

-       page 1 line 21: the nucleolus and not The nucleolus

Author Response

Manuscript ID: cells-544996

Type of manuscript: Review

Title: Nucleolar and Ribosomal DNA Structure Under Stress: Yeast Lessons for 
Aging and Cancer

Journal: Cells

(our reply in bold, next or below the corresponding reviewer’s comments)

REVIEWER #1

This review by Matos-Perdomo and Machin summarizes the functions played by the nucleolus in the context of aging and cancer, in particular based on work performed in yeast. The topic is extremely interesting and novel. The manuscript is very well-written and of broad interest.

The Authors provide important information about the various and complex level of alterations nucleolar function and integrity undergo, and this is a crucial (and yet mostly overlooked) aspect for the understanding of the functional consequences of nucleolar activity in physiological and pathophysiological systems.

We really appreciate comments from reviewer #1; they are very constructive and useful to further improve the quality of the manuscript.

I have a few concerns that will be important to address before publication to further improve the manuscript understanding for broader readerships:

1)   It is somehow difficult to follow the sequence of the different chapters. On page 2 line 39, it would be convenient to provide the summary of the way the review is structured.

We do agree in this aspect. We have included (on page 2, line 42 onwards) a summary of the review’s structure, in order to clarify the different issues addressed.

2)   It would be useful to clearly define from the beginning what is meant by longevity and what by ageing in yeast and in general in complex organisms.

Again, this is a good clarification, which we took for granted. We have included on the Introduction section, a definition of both terms.

3)   The Authors cite 229 references and this is certainly an immense effort. Nevertheless, in some cases it would be useful to describe the content a bit more in details, for example on page 4 related to refs from 48 to 52. The text is too compact and the content results difficult to follow.

We agree with reviewer’s comments on this issue. We acknowledge that the amount of references could be far too extensive. We were concerned about this and that is, in part, the reason we decided to leave out of this review important data about rDNA damage and the nucleolus.

As for the given example, we understand that the reviewer points out that it does not make much sense to mention the enlargement of the nuclear envelope at the location of the rDNA. However, we believe that feature must have an important impact on the nucleolar morphology, along with the rDNA size and liquid phase separation. We now explain this better in this chapter.

4)   The data shown in Figure 2 require a reference. Is this original work?

Figure 2 is original picture from our lab, not included in our previous publications. We have properly acknowledge the figure by stating that it is an original work as reviewer suggest.

5)   There is an imbalance between the aging and cancer part and a gap between the cancer part and the studies in yeasts. Also in the abstract it should be more specifically expressed what aspects of cancer biology are related to nucleolar stress response. Certainly p53 is a major link but it should be explained in more detail, for example by summarizing similarities and differences between yeast and higher eukaryotes in a table. Moreover how the work in yeast was helpful for cancer biology should be clarified more explicitly.

We appreciate this commentary, since we doubt about including the cancer section in the final submission of the manuscript. We believe the fact that we mention the NSR and p53 in the abstract already makes clear such connection, which we then develop further in the main text.

We have proceeded to include a table stating the differences in the nucleolus and the rDNA between yeast and higher eukaryotes.

As for how the yeast Saccharomyces cerevisiae has contributed to our understanding of cancer biology, we believe this is out of the scope of this article, which only focuses on the nucleolus. Just take into consideration how basic research in yeast has been critical to uncover the basic cellular mechanisms underlying the cell cycle, the checkpoints, the DNA damage response, accurate chromosome segregation, MAP kinases, Ras/cAMP, etc; all important aspects in cancer biology. The nucleolus is just another, often overlooked, player in this complex disease. 

Minor points:

-       abstract line 10 should be NSR and not NRS

We have changed this, thanks.

-       page 1 line 21: the nucleolus and not The nucleolus

We have changed this, thanks.

Reviewer 2 Report

The authors offer a review “Nucleolar and Ribosomal DNA Structure Under Stress: Yeast Lessons for Aging and Cancer”.

Major impression: The authors published in Cell Cycle 2018 their experimental work showing contraction of rDNA loop in the budding yeast under heat shock in TORC1-dependent manner.

Here they try to expand the story to cancer and aging in Metazoans. While concerning the yeast, the review is very interesting and useful,  for such complex problems in Metazoans as aging and cancer, some important issues are lacking and some views seem superficial or outdated.

Namely, (1) The relationship between mTORC1 pathway, DNA damage response (DDR), and genome instability.  We cannot consider aging without DDR, which is its most general hallmark. In this, the work from the lab of T. Kobayashi is extremely meaningful. Although many of their articles are cited, I would suggest to narrate more in detail the latest article Kobayashi, T. A new role of the rDNA and nucleolus in the nucleus - RDNA instability maintains genome integrity. BioEssays 2008, 30, 267–272. It would be nice to mention this author, who has contributed so much for deep understanding the problem, by surname at least once. In turn, both DDR and genome instability is a road to cancer and a hindrance to proliferation, too.

(2) mTORC1 is inhibiting autophagy Autophagy has not been even shortly observed. Autophagy (as well as TORC1 inhibitors) have dual complex effects in cancer (PMID: 29329237).  

In turn, for yeasts, both problems – DDR in rDNA and autophagy have been described in the phenomenon of piecemeal rDNA autophagy (PMID: 12529432), which is absent in this review.

Particular points.

2. The Nucleolus as a Marker of Cancer and Aging in Metazoans.

The authors operate mostly with the literature data on the nucleolar size and form – large and irregular in cancer – which is prognistically unfavourable and also large and fragmented – in senescence. It remains unclear why in the first case the cells are proliferating, while in the second, arresting proliferation. It is worth to mention with corresponding references that the number and size of nucleoli in cancer cells is associated not only with enhanced metabolism but even more, with polyploidy and aneuploidy, thus the latter (rDNA gene dosage multiplication) is a true case for the nucleolar cancer mark (Aneuploidy is the first hallmark of cancer).

3. The Structure of the Nucleolus and the rDNA in the Yeast Saccharomyces cerevisiae

It would be good, to give for the less informed reader before speaking on rDNA a simple morphological description :  In budding yeast, the nucleolus is a crescent-shaped structure occupying roughly one-third of the nuclear volume, abutting the nuclear envelope. 

3.1. Morphological Changes of the Yeast Nucleolus During the Cell Cycle.

p. 4, lines 106-107 rDNA is disorganized in G1the rDNA units are stained by the FISH probe as scattered foci within the nucleolar space [53]. The word “disorganized” should be substituted by ‘presented by disseminated clusters” or likewise.

6. Nucleolar Stress Remodelling and p53 Stabilization in Cancer

258 distinguished under the microscope: the fibrillar center (FC), the dense fibrillary center (DFC)

It is a misprint – should be “dense fibrillar component

The authors compare the impact on the nucleolus of the different sources of stresses – genotoxic and those caused by TOR inhibitors or similar which as they think may be less toxic but anti-proliferative, and possibly being safer. This is again a simplistic view on cancer.

Current knowledge cannot reduce anti-cancer strategy to antiproliferative one. Reprogramming of cancer cell is in the root of its resistance to extinction. In addition, TOR inhibitors can both activate and inhibit autophagy (depending on the context), while the latter can be both anti- and pro-tumorous.

305 a bad prognostic for non-small cells in lung cancer [216]. This is a nosological form of lung cancer “non-small-cell lung cancer”- should correct.

 The authors provide a good review of the p53-mdm2 metabolism and role of the nucleolus in it.

Still, it is strange not to mention that 60% of tumours have mutant TP53 and most of the rest inactivate it in some other way (Kastan MB Wild-type p53: tumors can't stand it. Cell 2007).

Moreover, the wrong nucleolar turnover of p53 in aging and progeria lead to nucleolar aggresomes – the issues which should be also mentioned speaking about stress-response functions of the nucleoli (.PMID:21425306; ) and in the association of nucleolar aggresomes with senescence, the circular nucleolar DNAs in metazoan aging and cancer (PMID: 28068183

In conclusion, the review may be useful for the readers of Cells but it is superficial concerning cancer and partly also aging in Metazoans. I suggest either to re-write the paragraphs concerning cancer of Metazoans in a more updated comprehensive way or reshape the review and do not go into this complex issue.  

Author Response

Manuscript ID: cells-544996

Type of manuscript: Review

Title: Nucleolar and Ribosomal DNA Structure Under Stress: Yeast Lessons for 
Aging and Cancer

Journal: Cells

(our reply in bold, next or below the corresponding reviewer’s comments)

REVIEWER #2 

The authors offer a review “Nucleolar and Ribosomal DNA Structure Under Stress: Yeast Lessons for Aging and Cancer”.

Major impression: The authors published in Cell Cycle 2018 their experimental work showing contraction of rDNA loop in the budding yeast under heat shock in TORC1-dependent manner.

Here they try to expand the story to cancer and aging in Metazoans. While concerning the yeast, the review is very interesting and useful, for such complex problems in Metazoans as aging and cancer, some important issues are lacking and some views seem superficial or outdated.

We do appreciate general comments as well as specific points outlined by reviewer #2, although we do not agree in certain points that we discuss below.

The review was intended to recompile all the information regarding to the structure of the nucleolus/rDNA, and how TORC1 inhibition impinges on it, and relate this to the actual knowledge of aging (mainly in yeast) and cancer in higher eukaryotes. We do agree with reviewer’s general comments about the lack of some issues that we also think are important. However, due to space limitation in other to keep the manuscript in a reasonable understandable way, we have not included them. Our take home message was that the nucleolar structure could be used as an indicator of longevity/aging (refs 29-35 and 179 of our manuscript) the same as is already used as a prognostic marker (among many others) in cancer by pathologists. We now stress this message in the MS.

Namely, (1) The relationship between mTORC1 pathway, DNA damage response (DDR), and genome instability.  We cannot consider aging without DDR, which is its most general hallmark. In this, the work from the lab of T. Kobayashi is extremely meaningful. Although many of their articles are cited, I would suggest to narrate more in detail the latest article Kobayashi, T. A new role of the rDNA and nucleolus in the nucleus - RDNA instability maintains genome integrity. BioEssays 2008, 30, 267–272. It would be nice to mention this author, who has contributed so much for deep understanding the problem, by surname at least once. In turn, both DDR and genome instability is a road to cancer and a hindrance to proliferation, too.

We totally agree with reviewer impression on this, the DNA Damage Response (DDR), represents an important aspect of both cancer and aging, and in our lab we have several publications on DDR itself, including our latest paper (Ayra-Plasencia & Machín. Nat Commun. 2019;10(1):2862). In addition, anaphase bridges are well known for triggering aneuploidy and genetic instability, as well as DNA double strand breaks upon cytokinesis completion; all important aspects of cancer development. Once again, we have decided not to include this topic on the manuscript, albeit we could have cited many of our own works related to rDNA anaphase bridges and partition of the nucleolus in late anaphase.

Although the DDR was not the topic and focus of the manuscript, we very briefly reference it on page 9, line 273 and 274 (now page 10, lines 309-311 references 202 & 203) and include a new reference on specific nucleolar DDR (reference 204) . We are considering writing up another review in this specific issue to be published elsewhere. Now, we state in the introduction we are not going over DDR and nucleolus in this review, while acknowledging its enormous importance.

We have centred the attention on the compaction and condensation of the rDNA/nucleolus and how different stresses, most of them well known TORC1 inhibitors, impinge on the yeast nucleolar structure. Certainly, the DDR is one important aspect of aging, but we disagree with reviewer #2 point about the DDR being the most important aspect for aging. We think, based on the scientific literature, that the DDR is as important as proteostasis, free radicals or telomere attrition. All these aspects represent what is known as the hallmarks of aging (reviewed in (now ref 244):  López-Otín et al, Cell 2013). Although genomic instability is put on top of the primary hallmarks (cause of damage), and the importance in aging research of, for example, accelerated aging diseases (defects in DNA repair) is of no doubt; the only interventions that have clearly extended lifespan and reduced aging throughout different taxa (yeast, worms, flies and mammals), are calorie restriction, TOR inhibition and sirtuins activity. Thus, although we recognize the important point reviewer #2 is mentioning, we thought that including a section on DDR, would further complicate and extend the scope of the manuscript.

Many theories of aging have been proposed: From the more general (mutation accumulation, disposable soma, antagonistic pleiotropy and free radical) to the more specific (DNA damage, loss of proteostasis, telomere shortening, rDNA, calorie restriction, aging causing damage versus damage causing aging, etc.); probably none of them are unique single factors as drivers of cancer and aging. Although we favour Kobayashis’s model or rDNA theory of aging, probably this operates through rDNA damage, replication, transcription, recombination, depletion of nucleolar factors, ribosome biogenesis, etc. In our opinion, we respectfully think that reviewer #2 has misunderstood the message of the manuscript; many other functions downstream of TOR (stress response, proteostasis, autophagy, mitochondrial function, genomic instability and replication stress, metabolic alterations, pH homeostasis, and so on), not only the ones covered in the manuscript, are also involved in the aging process. Remarkably, inhibiting TOR impinges on all those functions. We just concentrated on a  few aspects that we thought were important for what is known about rDNA/nucleolar condensation upon TOR inhibition and stress.

On the other hand, we do not certainly understand reviewer #2 comment on T. Kobayashi references. We truly appreciate and admire Kobayashi’s works; nonetheless, they have been extensively referenced throughout the whole manuscript, up to ten papers (in different references). Furthermore, we noticed the very recent article and model proposed for counting rDNA repeats by Kobayashi (now ref 45 on the manuscript). Moreover, the exact reference that reviewer is stating, “A new role of the rDNA and nucleolus in the nucleus - RDNA instability maintains genome integrity. BioEssays 2008, 30, 267–272”; is already appropriately referenced in the manuscript (now ref 241). In fact, we give record to a whole rDNA theory of aging by Kobayashi, including a more recent reference that further extends the ideas behind it (now ref 242: Ganley, A.R.; Kobayashi, T. Ribosomal DNA and cellular senescence: New evidence supporting the connection between rDNA and aging. FEMS Yeast Res 2014, 14, 49–59) and (now ref 37: Kobayashi, T. Ribosomal RNA gene repeats, their stability and cellular senescence. Proc. Jpn. Acad. Ser. B. Phys. Biol. Sci 2014, 90, 119–29).

Regarding the proposal to name Kobayashi within the main text, we must say we opted to write the manuscript in an impersonal way; only few names are given in the introduction, (including the Nobel Prize awarded B. McClintock), and none of them are alive but one. Having this in mind, other authors would also deserve to be equally mentioned; for example, TOR and TORC1 function: M. Hall, R. Loewith, C.K.Tsang and X.S. Zheng; Aging and sirtuins: L. Guarente and D. Sinclair; rDNA structure: V. Guacci and D. Koshland; rRNA processing: D.Tollervey, etc, just to mention a few of them. Anyhow, at reviewer’s request, we have specifically mentioned T. Kobayashi in the text when we mention his theory of aging at the end of the review.

(2) mTORC1 is inhibiting autophagy has not been even shortly observed. Autophagy (as well as TORC1 inhibitors) have dual complex effects in cancer (PMID: 29329237).  

In turn, for yeasts, both problems – DDR in rDNA and autophagy have been described in the phenomenon of piecemeal rDNA autophagy (PMID: 12529432), which is absent in this review.

Again, we agree with reviewer’ comment on this. Autophagy is an extremely important, yet complicated process, involved in aging and cancer progression. As reviewer said, dual aspects of mTOR inhibition are present in cancer biology. Although we briefly mention in the manuscript that TOR signalling inhibits autophagy (on page 5, line 156 references 90 and 91), describing this process is, in our opinion, outside of the main scope of the manuscript. Regarding to reference PMID: 12529432 that the reviewer suggests, in that paper what is shown by transmission electron microscopy are nucleolar portions included in piecemeal microautophagy nuclear (PMN) structures, but not the rDNA specifically. Furthermore, we included a reference (now ref 141) that states that rDNA and nucleolar components are selectively excluded and destined for nucleophagy degradation, respectively. (Golam Mostofa, M.; Rahman, M.A.; Koike, N.; Yeasmin, A.M.; Islam, N.;Waliullah, T.M.; Hosoyamada,S.; Shimobayashi, M.; Kobayashi, T.; Hall, M.N.; Ushimaru, T. CLIP and cohibin separate rDNA from nucleolar proteins destined for degradation by nucleophagy. J. Cell Biol 2018, 217, 2675–2690). The authors show in that work, that the rDNA is protected from the autophagic process, avoiding thus rDNA degradation. Even more, where nucleophagy occurs is still not known, and this is rarely observed in mammalian cells (Sorting the trash: Micronucleophagy gets selective J Cell Biol. 2018 Aug 6; 217(8): 2605–260; PMID: 30006460) (and now reference 142 of our manuscript). To further extend this point, in our Cell Cycle paper, PMID: 29166821, it can be observed how the nucleolar Nop1 signal is somehow displaced from the DAPI and rDNA (Cdc14) signals, under Heat Shock and rapamycin treatment in a small percentage of cells (figures 5A and 6A in our paper).

Particular points.

2. The Nucleolus as a Marker of Cancer and Aging in Metazoans.

The authors operate mostly with the literature data on the nucleolar size and form – large and irregular in cancer – which is prognistically unfavourable and also large and fragmented – in senescence. It remains unclear why in the first case the cells are proliferating, while in the second, arresting proliferation. It is worth to mention with corresponding references that the number and size of nucleoli in cancer cells is associated not only with enhanced metabolism but even more, with polyploidy and aneuploidy, thus the latter (rDNA gene dosage multiplication) is a true case for the nucleolar cancer mark (Aneuploidy is the first hallmark of cancer).

We appreciate this comment and agree with reviewer’s view on this. We cannot give a good explanation on why senescent cells present a fragmented nucleolus, and this is not the purpose of a revision of the scientific literature. The hypothesis behind is that, as we stated on (now) page 3, lines 84 to 87 (references 32-35), nucleolar fragmentation follows nucleolar enlargement in both yeast and senescent cell lines. Senescence is defined (on Biology of Aging; R. McDonald; Garland Science, Taylor & Francis 2014) as the age-related changes at the end of an organism’s life span that affect vitality and function, increase the likelihood of death, and are not directly related to disease. Senescent cells present several features: enlarged morphology, arrest in cell division and lengthening of the cell cycle, decrease in RNA synthesis and overall rate of protein synthesis, responsiveness to mitogenic signals, etc. Thus, senescent cells are still capable of responding to growth stimulation (by TOR for example) but not able to resume proliferation (PMID: 22394614, PMID: 18948731 and PMID: 19923900).  A feasible explanation based on this was recently published by Neurohr et al; Cell 2019 ( PMID: 30739799) where they show both in yeast and human fibroblasts, that excessive cell growth leads to cytoplasm dilution and senescence, that is, under a certain growth threshold, cells cannot scale up macromolecules with the corresponding increase in cell volume. Thus, as they state, “preventing excessive cell growth is important” and “growth beyond a cell type-specific ratio contributes to senescence” because contributes to loss of cell function. On the other hand, it is well known that cancer cells respond to growth stimulation and are able to proliferate, but not entering into the senescence programme (bypassing senescence) (PMID: 17700693). Cancer cells enter senescence, only by oncogene-induced senescence (OIS), that is, by disabling oncogene pathways or by inducing tumour-suppressor pathways, and by senescence-induced chemotherapy (PMID: 20029423). Thus, it could be possible that normal cells enlarge the nucleolus during aging up to senescence, arriving finally to an end point, where the nucleolus becomes fragmented, due to the uncoupling of macromolecular synthesis with cell volume and loss of nucleolar homeostasis and integrity.

We do agree with reviewer’s statement “number and size of nucleoli in cancer cells is associated not only with enhanced metabolism but even more, with polyploidy and aneuploidy, thus the latter (rDNA gene dosage multiplication) is a true case for the nucleolar cancer mark (Aneuploidy is the first hallmark of cancer).” We have included this on the manuscript. Actually, this was not included before only because the enlarged nucleolar phenotype was noticed by pathologists, long before the aneuploidy concept was established as the first hallmark of cancer. We thought of including references about rDNA copy number and tumour genetic context (e.g. PMID: 28880866) in the first instance, but we left it out in the last version under the assumption it would rather complicate the explanations. We include new references on this as the reviewer suggests. 

3. The Structure of the Nucleolus and the rDNA in the Yeast Saccharomyces cerevisiae

It would be good, to give for the less informed reader before speaking on rDNA a simple morphological description :  In budding yeast, the nucleolus is a crescent-shaped structure occupying roughly one-third of the nuclear volume, abutting the nuclear envelope. 

We have included the paragraph about rDNA structure as the reviewer suggests. 

3.1. Morphological Changes of the Yeast Nucleolus During the Cell Cycle.

p. 4, lines 106-107 rDNA is disorganized in G1the rDNA units are stained by the FISH probe as scattered foci within the nucleolar space [53]. The word “disorganized” should be substituted by ‘presented by disseminated clusters” or likewise.

We have changed this, as the reviewer wisely suggests.

6. Nucleolar Stress Remodelling and p53 Stabilization in Cancer

258 distinguished under the microscope: the fibrillar center (FC), the dense fibrillary center(DFC) 

It is a misprint – should be “dense fibrillar component

We acknowledge the misprint and changed it. 

The authors compare the impact on the nucleolus of the different sources of stresses – genotoxic and those caused by TOR inhibitors or similar which as they think may be less toxic but anti-proliferative, and possibly being safer. This is again a simplistic view on cancer.

Current knowledge cannot reduce anti-cancer strategy to antiproliferative one. Reprogramming of cancer cell is in the root of its resistance to extinction. In addition, TOR inhibitors can both activate and inhibit autophagy (depending on the context), while the latter can be both anti- and pro-tumorous.

We do not intend in this review to foresee what is the best approach against cancer in the following years. We only speculate that inhibition of rRNA transcription/processing should be explored as an alternative to more toxic antiproliferative strategies. We agree that it is likely that future antitumor therapy will not be reduced to antiproliferative actions. However, antiproliferation is a well-proven strategy and still the core of antitumor chemotherapy in the clinic. At present, there are multiple clinical trials with new antiproliferative drugs whose mode of action is similar to what clinicians already employ, but which are supposed to be less clastogenic or aneugenic (e.g., kinase, kinesin or TOPO2alpha catalytic inhibitors, etc.)

Incidentally, we have just noticed that the referred text mentioned “cytotoxicity” when we actually meant “genotoxicity”.

305 a bad prognostic for non-small cells in lung cancer [216]. This is a nosological form of lung cancer “non-small-cell lung cancer”- should correct.

We have corrected this, thanks. 

The authors provide a good review of the p53-mdm2 metabolism and role of the nucleolus in it.

Still, it is strange not to mention that 60% of tumours have mutant TP53 and most of the rest inactivate it in some other way (Kastan MB Wild-type p53: tumors can't stand it. Cell 2007).

Moreover, the wrong nucleolar turnover of p53 in aging and progeria lead to nucleolar aggresomes – the issues which should be also mentioned speaking about stress-response functions of the nucleoli (.PMID:21425306; ) and in the association of nucleolar aggresomes with senescence, the circular nucleolar DNAs in metazoan aging and cancer (PMID: 28068183

We draw the attention to the existence of both p53 dependent and independent nucleolar stress responses (now ref 225), though we have included the corresponding references that reviewer suggests, as well as others stating p53 transcriptional variants influencing tumour progression.

In conclusion, the review may be useful for the readers of Cells but it is superficial concerning cancer and partly also aging in Metazoans. I suggest either to re-write the paragraphs concerning cancer of Metazoans in a more updated comprehensive way or reshape the review and do not go into this complex issue

We understand that we cannot cover everything in this review. It already contains more than 200 references. As stated above, we intended to focus on nucleolar/rDNA changes in yeast upon cell stress, mostly as an acute response (e.g., not long-term effects of DNA damage) and how this might relate to aging and cancer. It is true that aging has many faces, especially in metazoans, and the nucleolus is just one of them. As for cancer, yeast cells are always proliferating under favourable conditions. Thus, it is complex to correlate yeast physiology to human cell physiology. However, the basis for an effective proliferation are highly conserved between yeast and humans, being TOR one of the most representative examples. That is the reason we decided to cover both issues in this review. What we have now done is to explain what we talk about and what we do not at the beginning of the review, so readers can appreciate this better.